# Characterization of the Glucan-Branching Enzyme GlgB Gene from Swine Intestinal Bacteria

**DOI:** 10.3390/molecules28041881

**Published:** 2023-02-16

**Authors:** Yuqi Shao, Weilan Wang, Ying Hu, Michael G. Gänzle

**Affiliations:** Department of Agricultural, Food and Nutritional Science, University of Alberta, Edmonton, AB T6G 2P5, Canada

**Keywords:** branching enzyme, GlgB, intestinal microbiome, colonic starch fermentation, resistant starch, starch digestibility

## Abstract

Starch hydrolysis by gut microbiota involves a diverse range of different enzymatic activities. Glucan-branching enzyme GlgB was identified as the most abundant glycosidase in Firmicutes in the swine intestine. GlgB converts α-(1→4)-linked amylose to form α-(1→4,6) branching points. This study aimed to characterize GlgB cloned from a swine intestinal metagenome and to investigate its potential role in formation of α-(1→4,6)-branched α-glucans from starch. The branching activity of purified GlgB was determined with six different starches and pure amylose by quantification of amylose after treatment. GlgB reduced the amylose content of all 6 starches and amylose by more than 85% and displayed a higher preference towards amylose. The observed activity on raw starch indicated a potential role in the primary starch degradation in the large intestine as an enzyme that solubilizes amylose. The oligosaccharide profile showed an increased concentration of oligosaccharide introduced by GlgB that is not hydrolyzed by intestinal enzymes. This corresponded to a reduced in vitro starch digestibility when compared to untreated starch. The study improves our understanding of colonic starch fermentation and may allow starch conversion to produce food products with reduced digestibility and improved quality.

## 1. Introduction

Dietary polysaccharides are a major source of energy for intestinal microorganisms in the gut of monogastric animals [1]. Of the dietary carbohydrates, most sugars and starch are digested in the small intestine, while the main body of non-digestible oligo- and polysaccharides and insoluble particles escape digestion by host enzymes and are fermented in the terminal ileum or the colon [2]. Those non-digestible carbohydrates provide an essential substrate for gut microbiota, and the end products, short-chain fatty acids (SCFA), confer health benefits to the host [3,4]. Resistant starch is the non-digestible fraction of starch that is resistant to small intestinal digestion by salivary and pancreatic α-amylases and reaches the large intestine. Resistant starch was estimated to be the most abundant non-digestible carbohydrate in most human diets and the largest source of energy for colonic fermentation [2,5]. Resistant starch occurs in multiple forms; type 3 resistant starch includes retrograded starch and crystalline amylose, which escapes hydrolysis by pancreatic enzymes owing to its low solubility [6,7]. In the metagenomes of monogastric animals, enzymes for starch hydrolysis are among the most abundant carbohydrate-active enzymes (CAZymes) [8]. Starch utilization by gut microbiota in pigs and humans requires a diverse range of carbohydrate-active enzymes expressed by different microbial members [9]. The mechanism of starch utilization was first studied in *Bacteroides thetaiotaomicron* [10]. In general, the first step in the starch degradation process occurred by binding of the starch molecules to the cell surface with starch binding and transport proteins [11]. A portion of starch is hydrolyzed to oligosaccharides by extracellular amylases prior to periplasmic and subsequent cytoplasmic hydrolysis [1,10,11,12]. In the human intestine, *Ruminococcus bromii*, which releases oligosaccharides during the hydrolysis of resistant starch, facilitates the growth of other amylolytic bacteria and is, therefore, considered to be a keystone species for the metabolism of resistant starch [13].

Starch hydrolysis by intestinal bacteria is carried out predominantly by extracellular or periplasmic glycosyl hydrolase (GH) family 13 amylases or (neo-)pullulanases, which collectively hydrolyze α-(1→4)-, α-(1→6)- or α-(1→4,6)-linked α-glucans, including amylose and amylopectin [1]. A metagenomic study identified an α-glucan branching enzyme from GH family 13 that is potentially involved in starch metabolism in the intestine of swine [8]. Branching enzymes were also abundant in humans, chicken and cow metagenomes, and the sequence diversity of gene coding for the branching enzymes differentiates between hosts, with the highest similarity in pigs and humans [14]. Glucan-branching enzymes add α-(1→4,6) branching points to α-(1→4)-linked linear glucan chains [15]. These branching points reduce starch digestibility by pancreatic amylases [16,17], but the formation of highly branched and soluble structures may facilitate colonic fermentation of starch, particularly crystalline amylose, by increasing their accessibility to bacterial glycosyl hydrolases [18].

To date, only a few branching enzymes from intestinal bacteria have been biochemically characterized [18,19]. It was, therefore, the aim of this study to biochemically characterize the gene product of *glgB* coding for a glucan-branching enzyme found in swine intestine and to investigate its role in the formation of α-(1→4,6)-branched oligosaccharides from starch. In this study, *glgB* was cloned and over-expressed in *E. coli* to investigate the branching activity of the purified glucan-branching enzyme GlgB on amylose and starch and to elucidate its role in colonic starch utilization.

## 2. Results

### 2.1. Cloning of Glgb from Swine Intestinal Bacteria in E. coli and Protein Purification

Initially, the presence of *glgB* in swine intestinal microbiota was predicted based on the annotation of metagenomic-assembled genomes. The *glgB* sequence was cloned from an uncharacterized member of the phylum *Firmicutes* [8]. The *glgB* sequences were verified by PCR amplification of 1941 bp amplicons and sequencing (Figure 1). The insertion of *glgB* in pET-28b+ was also verified by PCR amplification. GlgB was purified by his-tag affinity chromatography after over-expression in *E. coli* BL21 Star (DE3). Purification of GlgB with a predicted molecular weight of 72 kDA was verified by SDS-PAGE (Figure 2). The concentration of purified GlgB was 1 g/L, as determined with bovine serum albumin as standard.

### 2.2. Quantitative Determination of Iodine-Binding Amylose

Amylose binds iodine to form a stable blue complex. The activity of GlgB was initially assessed by quantification of iodine-binding amylose, before and after incubation with GlgB, with six different starches and pure amylose as substrates [18]. The purified GlgB displayed activity both on amylose and starch and reduced the amylose content of all 6 starches and amylose by more than 85% (Table 1). The reduction of the amylose content was 97% with amylose as substrate (Table 1). The starches with a higher amylose content, namely, fava bean, potato and pea starch, also tended to have a high reduction of the amylose content of 96.9%, 96.6%, and 96.0%, respectively, after treatment with GlgB. Corn, wheat and barley starches were characterized by a lower initial amylose content, and the reduction of the amylose content after GlgB treatment was less than 90% (Table 1). The activity of GlgB was also assessed with raw starches without gelatinization, including corn, pea and wheat, by quantification of iodine-binding amylose before and after incubation with GlgB. The purified GlgB also displayed activity on raw starches (Appendix A), with an average of 18% reduction of the amylose content after treatment with GlgB. GlgB enzymatic treatment, thus, showed a smaller impact on raw starches than on gelatinized starches.

### 2.3. Oligosaccharide Profile after Amylase Hydrolysis

To characterize the conversion products after GlgB treatment, GlgB-treated starches and amylose were hydrolyzed with α- and β-amylases to hydrolyze the linear α-(1→4)-chains, but not the α-(1→4,6) branching points. The resulting oligosaccharide profiles after hydrolysis were analyzed using HPAEC-PAD (Figure 3, Appendix A). Starches and amylose without GlgB treatment served as controls. Untreated amylose had fewer peaks corresponding to amylase-resistant oligosaccharides when compared to potato starch (Figure 3). In both samples, several oligosaccharide peaks eluting between 28 and 45 min were observed after treatment with GlgB. This elution time corresponds to linear α-glucooligosaccharides with α-(1→4) and α-(1→6) linkages and a degree of polymerization of more than 10 [20]. The peak pattern, thus, demonstrates that GlgB converts amylose and starch by introducing linkage types that are not hydrolyzed by amylases. The peak patterns of amylase-resistant oligosaccharides after treatment with GlgB were comparable between amylose and potato starch (Figure 3). The same pattern was also observed with starches from pea, fava bean, corn, wheat and barley (Appendix A). The comparison of oligosaccharide patterns obtained from treated and untreated amylose (Figure 3A) and from treated and untreated starches indicate that some of the amylase-resistant oligosaccharides introduced by GlgB are identical to those present in amylopectin, for example, the peaks eluting at ca. 25 min (Figure 3 and Appendix A). Peaks that eluted after 30 min were not present in untreated amylopectin (Figure 3B and Appendix A).

### 2.4. Molecular Size Distribution of GlgB-Treated Starch

The size distribution of GlgB-treated starch was analyzed using HPSEC (Figure 4), with a linear range of about 10,000 to 1,000,000 relative molecular weight. Potato and wheat starches were selected as the representative starches based on their different content of amylose to amylopectin. For both potato and wheat starch, GlgB treatment substantially reduced the concentration of high-molecular-weight molecules eluting at or close to the void volume of the column (5 mL) and increased the area of peaks with an elution volume between 10 to 20 mL (Figure 4). The molecular size of GlgB-treated wheat and potato starch differed (Figure 4), indicating that starch with different properties impacted GlgB activity, which led to different size distribution before and after GlgB treatment.

### 2.5. Quantitative Determination of Reducing Ends after Debranching

To directly determine the branching activity of GlgB, reducing ends were quantified after the debranching of amylose and starches with isoamylase and pullulanase. The concentration of reducing ends was expressed in µM/g of starch (Table 2). The concentration of reducing ends significantly increased amylose, fava bean starch and corn starch after GlgB treatment. After treatment with GlgB, the number of reducing ends was consistently about 250 µM/g starch, irrespective of the number of reducing ends of the substrates (Table 2). This suggested that the glucan-branching enzyme GlgB saturates at a certain amount of branching points, and it confirms the preference of GlgB for amylose as substrate. The number of branching points at saturation corresponds to approximately 0.04 mol reducing ends per mol glucose or 1 α-(1→4,6) branching point per 24 α-(1→4)-linked glucose moieties.

### 2.6. In Vitro Digestion of GlgB-Treated Starch Products

The digestibility of GlgB-treated starch products was determined by measuring the amount of glucose released after digestion of the starch products with pancreatic amylases and brush border enzymes. The concentration of glucose released in the GlgB-treated and untreated starch samples were compared (Table 3). GlgB treatment significantly (*p* < 0.05) reduced the glucose release for pea, fava bean and wheat starches. GlgB enzymatic treatment, thus, showed a moderate impact on the in vitro starch digestibility.

## 3. Discussion

Intestinal bacterial communities in swine and other monogastric animals harbor an extensive metabolic potential for starch utilization, but the amylolytic system has only been fully characterized in a few species [2]. The glucan-branching enzyme GlgB, which frequently occurs in genomes of *Firmicutes*, was identified as the most abundant glycosidase in swine intestine [8]. In this study, we cloned *glgB* from community DNA isolated from swine feces and expressed the gene in *E. coli* to characterize the enzyme activity on amylose and starches of different botanical origin. The branching activity of GlgB on amylose and starch was assessed by quantifying the reducing ends and by oligosaccharides analysis after hydrolysis with amylases, as well as an in vitro digestibility test, which provide us insight into its potential role for starch utilization in swine large intestine and in the development of functional foods.

### 3.1. Biochemical Characteristics of Glucan-Branching Enzyme GlgB

Glucan-branching enzymes were predominantly studied from thermophilic microorganisms or hyper-thermophilic microorganisms; a commercially available branching enzyme is also derived from a hyper-thermophilic bacterium [21,22,23,24,25,26,27,28]. Those glucan-branching enzymes share the same mechanism of enzyme action, but their biochemical characteristics differ from GlgB with respect to the optimum conditions for activity and the substrate specificity. Only a few studies isolated and characterized branching enzymes from intestinal bacteria. The first intestinal glucan-branching enzyme *glgB* gene was isolated from *Butyrivibrio fibrisolvens* [18]. The amino acid sequence of GlgB in this study is 60.92% identical to the glucan-branching enzyme GlgB (EC2.4.1.18) of *Butyrivibrio fibrisolvens*. The two enzymes share similar optimum conditions for enzymatic reaction, and both showed activity on amylose [18]. Li et al. provide a detailed biochemical characterization of a glucan-branching enzyme isolated from *Bifidobacterium longum* [19]. However, none of these previous studies investigated the substrate preference of GlgB. GlgB characterized in the present study displayed activity on both amylose and starch, but the highest enzymatic activity was discovered on amylose. Starch is composed of amylose and amylopectin; the amylose and amylopectin ratio vary, dependent on the source [29]. When comparing starches of different botanical origin, GlgB appeared to be more active on starches with a high amylose content, confirming a higher preference of this branching enzyme on amylose as the substrate. This higher preference on amylose was also confirmed by the number of reducing ends increased as a direct measure of branching activity. Only amylose showed a substantial increase in the number of reducing ends after GlgB treatment. This higher preference of GlgB towards amylose contrasts with the commercially available branching enzyme derived from *Rhodothermus obamensis*, which preferred branched amylopectin as its acceptor substrate [25,30]. In addition, the oligosaccharide profiles after amylase hydrolysis also showed a comparable pattern between amylose and starch, indicating that those branching points introduced by the GlgB were predominately from amylose. Therefore, GlgB is predominantly an amylose active enzyme. The botanical origin of starches also impacted GlgB activity.

### 3.2. Physiological Function of Glucan-Branching Enzyme GlgB in Colonic Starch Digestion

The metagenomic assembled genome encoding for GlgB was identified as a member of *Ruminococcaceae* based on its 16S rRNA sequence [8]. GlgB does not have a signal peptide to direct the protein for export from cytoplasm [8]; however, cloning from cloning of GlgB from *B. fibrisolvens* in *E. coli* revealed extracellular starch-clearing activity [18]. In addition, re-analysis of 50 MAG operon members of the Firmicutes with Operon Mapper (https://biocomputo.ibt.unam.mx/operon_mapper/, accessed on 27 January 2023) revealed that GlgB homologues are found in 6 different types of operons (Appendix A). In two of those, GlgB is associated with glycogen synthase and ADP-pyrophosphorylase, as expected for an enzyme with a role in glycogen synthesis. In four types of operons, however, *glgB* occurs as a stand-alone gene, in association with enzymes that do not relate to either glycogen synthesis or starch hydrolysis, or in operons that are organized as carbohydrate utilization loci (Appendix A). A putative physiological function of GlgB in colonic starch digestion can also be inferred based on its biochemical properties (Figure 5).

As the most abundant glycosyl hydrolase-family enzyme from *Firmicutes* [8], the glucan-branching enzyme GlgB displayed high activity on gelatinized amylose and starches. However, moderate activity on raw starches was also observed, suggesting the potential role of this GlgB as an enzyme that solubilizes amylose. Crystalline amylose is a type 3 resistant starch that escapes small intestinal digestion due to its low solubility and enters the colon as one of the main substrates for starch fermentation [6,7]. The observed activity of the glucan-branching enzyme on amylose and raw starches converts crystalline amylose into branched structure that is soluble and easier to access by other amylolytic enzymes. Therefore, colonic starch degradation is likely initiated by GlgB-mediated solubilization of crystalline amylose, followed by hydrolysis by other extracellular or periplasmatic amylolytic enzymes. In the *Bacteroides* amylolytic system, SusG is the sole extracellular amylase responsible for degrading starch into oligosaccharides molecules that can be transported through SusC into the periplasm, where most of the starch-degrading activity occurs with the periplasmic amylolytic enzymes Sus A and B (Figure 5) [1,2,10,11,12,31,32]. The role of glucan-branching enzyme GlgB in forming branched structure from crystalline amylose likely increases its accessibility to SusG by converting unbranched raw amylose substrates into highly branched structures with higher solubility. In addition, a reduced molecular weight after the GlgB treatment was also observed, which also facilitated the accessibility of raw amylose to allow hydrolysis by other intestinal enzymes. This physiological function of the glucan-branching enzyme confirms the hypothesis that the branching enzyme increased the solubility of the unbranched substrate [18]. Confirmation of a role of GlgB in starch hydrolysis requires, however, experimentation with an intestinal isolate that produces the enzyme.

### 3.3. The Use of Glucan-Branching Enzyme GlgB for Starch Conversion in Food Applications

Starch digestibility is influenced by the linkage type. Pancreatic amylases hydrolyze only α-(1→4) linkages; however, brush border glycosyl hydrolases act slower on α-(1→6) linkages, when compared to α-(1→4) linkages, and slower on α-(1→4,6) branching points than on α-(1→6)- or α-(1→4)-linked linear oligosaccharides [16,17,25]. Highly branched α-glucans treated with branching enzyme showed a remarkably slower digesting property both in vitro and in vivo, which was primarily due to the higher proportion of α-(1→4,6) branching points introduced [33]. A reduced in vitro starch digestibility was observed in this study after the GlgB treatment also indicated a slower digesting property of the GlgB-treated starch. Therefore, the potential of branching enzyme to form branched structures makes it highly suitable in functional food development, as well as quality improvement, especially in the baking industry [34]. The in vitro digestibility suggested a moderate impact of GlgB on starch. However, changes in bread texture and staling could be possible if the branching enzyme was incorporated [35,36].

In the baking industry, the nutritional and physical properties of bread are largely related to the physicochemical properties of starch [37]. High amylose content starch was used in order to increase the resistant starch levels in foods for enhanced nutritional value [38]. However, the use of high amylose flour in baking applications also significantly reduced bread volume and increased crumb firmness [39,40]. To make bread with both improved nutritional and physical properties, enzymatic starch modification is an alternative approach [34,37]. The addition of branching enzyme in wheat breadmaking resulted in an increased bread volume of 26% and decreased crumb firmness of 38% [36]. Branching enzyme treatment also retarded retrogradation of corn starch, corresponding to a slowed hardness increase [35]. Therefore, the use of glucan-branching enzyme GlgB in baking can be a promising solution to make bread with both improved nutritional value and quality. Further studies on the use of GlgB in bread baking can be a possible approach to investigate its role in functional food development.

In conclusion, the successfully cloned glucan-branching enzyme GlgB from swine intestinal bacteria showed strong branching activity, which forms branched oligosaccharides that are not hydrolyzed by intestinal enzymes. The biochemical characteristics of this branching enzyme indicate a higher preference towards amylose as its substrate, which further indicates a potential role in colonic starch digestion as an amylose solubilization enzyme. Those findings improved our understanding of colonic starch fermentation and may also allow starch conversion to produce bread with reduced digestibility while remaining higher quality.

## 4. Materials and Methods

### 4.1. Materials

Starches were chosen to represent a diverse botanical origin and a range of amylose contents of 30–70%. Amylose from potato and starches from potato and wheat were purchased from Sigma Aldrich (St. Louis, MO, USA). Starches from pea, fava bean and barley with over 95% purity were kindly provided by Thava Vasanthan; these starches were obtained by aqueous extraction from seeds obtained from Tomtene Seed Farms, Birch Hills, Saskatchewan, Canada. The commercial amylases Termamyl (α-amylase) and Fungamyl (fungal maltogenic alpha-amylase) were provided by BioNeutra Inc. (Edmonton, AB, Canada). Isoamylase HP (240 U/mg, 500 U/mL) and pullulanase M1 (30 U/mg, 650 U/mL) were purchased from Megazyme (Wicklow, Ireland).

### 4.2. Bacteria, Plasmid and Growth Condition

*Escherichia coli* BL21 Star (DE3) (Invitrogen, Waltham, MA, USA) with pET-28b+ (Novagen, Etobicoke, ON, Canada) was cultivated aerobically at 37 °C in Luria-Bertani (LB) broth containing 0.05 g/L kanamycin.

### 4.3. Cloning of Glgb in E. coli

Homologues of starch branching enzymes were identified in metagenomic assembled genomes by protein blast [8]. Two protein sequences (Genebank Accession numbers OP096417 and OP096418) with an amino acid identity of 60.92% to the α-(1→4)-glucan-branching enzyme GlgB (EC2.4.1.18) of *Butyrivibrio fibrisolvens* [18] were selected. The *glgB* genes were amplified from intestinal community DNA [8] with high-fidelity Tag DNA polymerase (Platinum^TM^ Taq DNA Polymerase High Fidelity, Invitrogen), with primers (5′ to 3′) B34F-SacI (GCTGAGCTCATGACAACTGTAGAAAAGAAA) and B34R-SalI (GAAGTCGACGAATTCAAATACCGCAACG), using DNA isolated from fecal samples [8] as template. Purified PCR amplicons and the pET-28b+ vector were digested with SacI/SalI and then ligated into recombinant plasmids pET-28b + *glgB.* Chemically competent cells of *E. coli* BL21 Star (DE3) (Invitrogen, Carlsbad, CA, USA) were transformed with the recombinant plasmids, according to the One Shot BL21 (DE3) Competent Cell Manual (Invitrogen, MA, USA). Transformants were screened with the primers B34F-SacI and B34R-SalI to identify clones harboring *glgB*, and the sequence of the insert was verified by Sanger-sequencing using the T7 primers 5′-TAATACGACTCACTATAGGG-3′ (University of Alberta, Faculty of Science, Canada). Clones B342 and B344 share the same nucleotide sequence (GeneBank accession numbers OP096417 and OP096418, respectively), and clone B342 was selected for all subsequent experiments.

### 4.4. Overexpression of GlgB and Protein Purification

Overexpression of GlgB in *E. coli* was induced by addition of 0.2 mM isopropyl β-D-1-thiogalactopyranoside (IPTG) to exponentially growing cultures in LB broth, corresponding to an optical density (OD_600nm_) of about 0.4, followed by overnight incubation at 25 °C. Cells were harvested by centrifugation at 5000 rpm for 10 min and disrupted with a bead beater for 20 s for 8 times. GlgB was purified using the HisPur^TM^ Ni-NTA Resin (Fisher Scientific, Loughborough, UK), concentrated using an Amicon^®^ centrifugal filter with 30 kDa molecular weight cut off (Millipore, Burlington, MA, USA). Purification was verified by sodium dodecyl sulfate polyacrylamide gel electrophoresis (SDS-PAGE). Protein concentration was measured by Bio-Rad protein assay dye reagent (Bio-Rad, Mississauga, ON, Canada), with bovine serum albumin as the standard.

### 4.5. Determination of GlgB Activity

Amylose from potato and starch from pea, fava bean, potato, corn, wheat and barley were used as substrates. The enzyme activity of GlgB was determined by measuring the reduction of the iodine-binding amylose [18], with modifications. Amylose and starches (10 g/L) were dissolved in 1 M sodium hydroxide and 50 mM sodium phosphate buffer and heated at 85 °C for 10 min, followed by adjusting the pH to 7.4 using 1 M HCl. GlgB was added into the reaction mixture to a concentration of 200 mg/L, and the enzymatic reaction was incubated at 37 °C for 24 h. The quantitative detection of iodine-binding amylose was conducted based on the liner correlation between iodine-binding amylose content and the corresponding absorption of amylose–iodine complex at 620 nm wavelength. After enzymatic reaction, samples (10 µL) were taken into the 96-well plate and mixed with 200 µL 0.2% iodine solution, and the absorbance at 620 nm wavelength was measured. The amylose concentration was determined with a standard curve, generated with amylose concentrations ranging from 0 to 8 g/L.

### 4.6. Analysis of Oligosaccharides Profiles by HPAEC-PAD

To determine the presence of branching points that are resistant to hydrolysis of linear α-(1→4)-chains by amylase, GlgB-treated amylose and starches were incubated with 0.02% (*v*/*v*) α-amylase at 95 °C and pH 5–6 for 1.5 h, followed by incubation with 0.02% β-amylase at 55 °C for 24 h. The oligosaccharides profiles after amylase hydrolysis were analyzed using high-performance anion-exchange chromatography, coupled to pulsed amperometric detection (HPAEC-PAD) on a Dionex ICS-3000 Ion Chromatography System (Dionex, Oakville, ON, Canada). The samples were separated on a Carbopac PA20, coupled to an ED40 chemical detector (Dionex, Oakville, ON, Canada). Water (A), 0.2 M NaOH (B) and 1 M NaOAc (C) were used as eluents, and samples were eluted at a 0.25 mL/min flow rate with the following gradients: 0 min, 68.3% A, 30.4% B and 1.3% C; 30 min, 54.6% A, 30.4% B and 15.0% C; 50 min, 46.6% A, 30.4% B and 23% C; 95 min, 33.3% A, 30.4% B and 36.3% C; followed by re-equilibration. Glucose, fructose, isomaltose and maltose were used as standards. Untreated amylose and starches served as controls.

### 4.7. Analysis of Molecular Size Distribution by HPSEC-RI

The molecular size distribution of starch samples after GlgB treatment was determined using high-performance size-exclusion chromatography (HPSEC) on an Agilent 1200 HPLC system (Agilent Technologies, Santa Clara, CA, USA), coupled to a refractive index (RI) detector. Starches were separated on a Superdex 200 Increase 10/300 GL column (GE Healthcare, Chicago, IL, USA) that was eluted with 18 MΩ water at 0.2 mL/min. Untreated starches served as controls.

### 4.8. Determination of GlgB Branching Activity

The branching activity of GlgB was also determined by quantification of the branching points in GlgB-treated and untreated starches. Branching points were quantified by quantification of reducing ends after the debranching of amylose and starches with isoamylase and pullulanase. In brief, after GlgB treatment, the pH of amylose and starch solutions was adjusted to 4.5 using 1 M hydrochloride acid, and samples were incubated with 23 mg/L isoamylase and 241 mg/L pullulanase at 40 °C for 24 h. Reducing ends after debranching were quantified by quantification of reducing ends with bicinchoninic acid [41], with modifications. After debranching, samples were mixed with a 2,2′-bicinchoninic acid solution with a 1:4 ratio and incubated at 75 °C for 30 min. The absorbance at 560 nm wavelength was measured after cooling. The concentration of reducing ends concentration was calculated with a standard curve generated with glucose. Data were expressed as µM reducing ends/g starch.

### 4.9. In Vitro Digestibility Assay

The in vitro digestibility of starch samples with and without the GlgB treatment was quantified using pancreatic and intestinal brush border enzymes [20]. Starch samples (1 mL, 10 g/L) were mixed with 1 mL of 50 mM sodium maleate buffer (pH 6.0), containing 10 g/L intestinal acetone powder from rat (Sigma-Aldrich) and 0.07 g pancreatin from porcine pancreas (Sigma-Aldrich). Three to five glass beads with 5 mm diameter were added, and samples were incubated at 37 °C for 4 h with agitation at 200 rpm. The reaction was stopped by heating the reaction mixture at 90 °C for 5 min. The samples were cooled and centrifuged for 3 min at 5000× *g*, and the glucose concentration was quantified using the glucose oxidase kit (Megazyme, Wicklow, Ireland).

### 4.10. Statistical Analysis

Data analysis of the reducing ends and in vitro digestibility assay was performed using paired *t*-test. *p* < 0.05 was considered statistically significant. Results are expressed as averages ± standard deviation or as representative chromatograms.

## Figures and Tables

**Figure 1 molecules-28-01881-f001:**
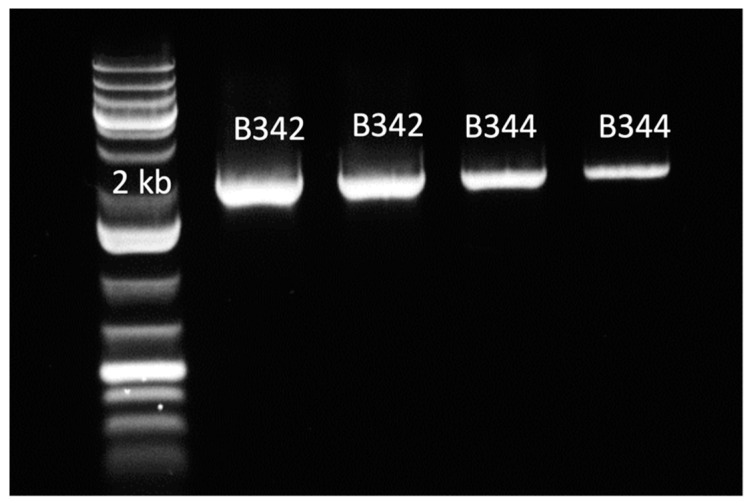
Verification of PCR amplification and insertion of *glgB* genes in plasmid DNA isolated from *E. coli* clones B342 and B344. Lane 1: DNA ladder; Lanes 2–3: *E. coli* clone B342; Lanes 4–5: *E. coli* clone B344.

**Figure 2 molecules-28-01881-f002:**
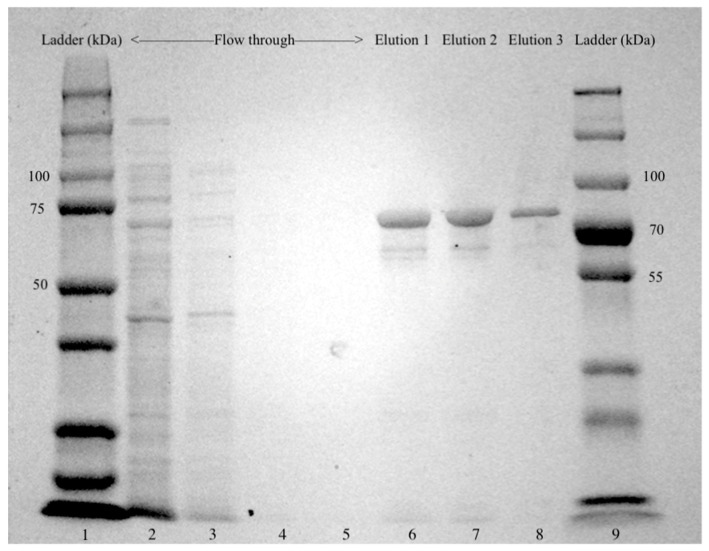
SDS-PAGE analysis of glucan-branching enzyme GlgB expressed in *E. coli* BL21 Star^TM^ (DE3). Crude cellular extract was loaded on a HisPur^TM^ Ni-NTA Resin and washed with PBS buffer containing 25 mM imidazole, and GlgB was eluted with PBS buffer containing 250 mM imidazole. Lanes 1 and 9: protein ladders; Lanes 2–5: Flow-throughs of crude cellular extract; Lanes 6–8: the 1st, 2nd and 3rd elution of GlgB.

**Figure 3 molecules-28-01881-f003:**
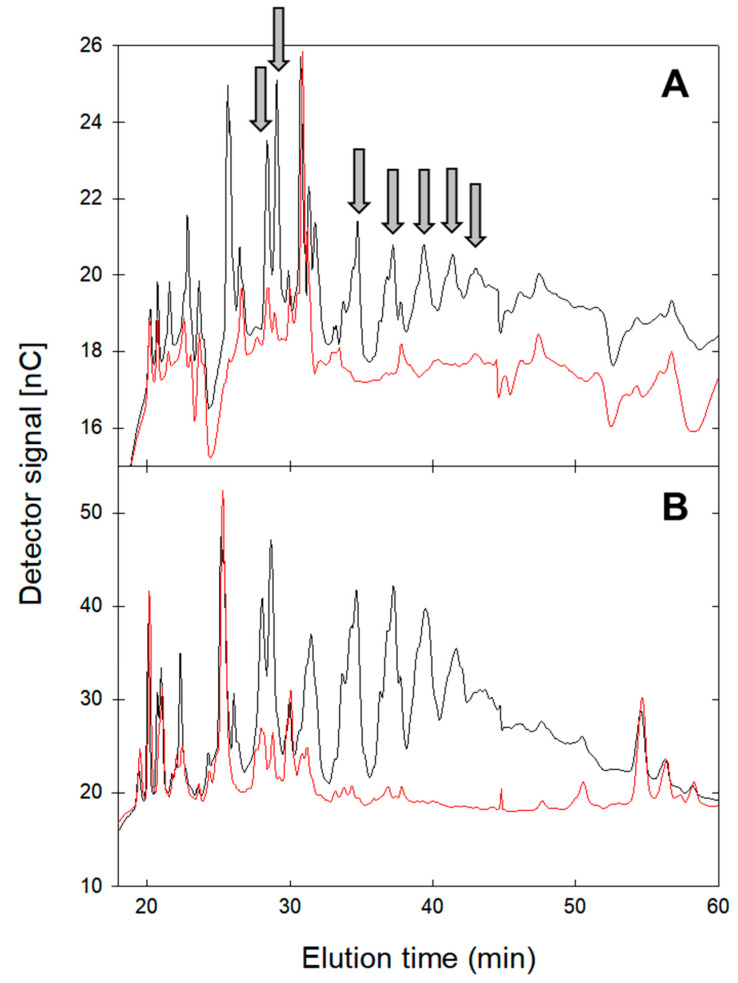
HPAEC-PAD profiles of oligosaccharides after hydrolysis of amylose (**A**) and potato starch (**B**) by α- and β-amylases. Amylose and potato starch were treated with GlgB (black line); untreated amylose and potato starch served as control (red line). Oligosaccharides that were present after GlgB treatment in treated amylose and potato starch are indicated with an arrow. Monosaccharides eluting between 3 and 10 min are not shown.

**Figure 4 molecules-28-01881-f004:**
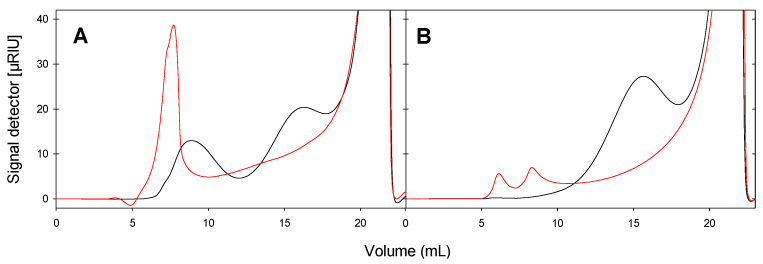
Size distribution of potato starch (**A**) and wheat starch (**B**) analyzed by HPSEC-RI. Starches were treated with GlgB (black line) or not (red line).

**Figure 5 molecules-28-01881-f005:**
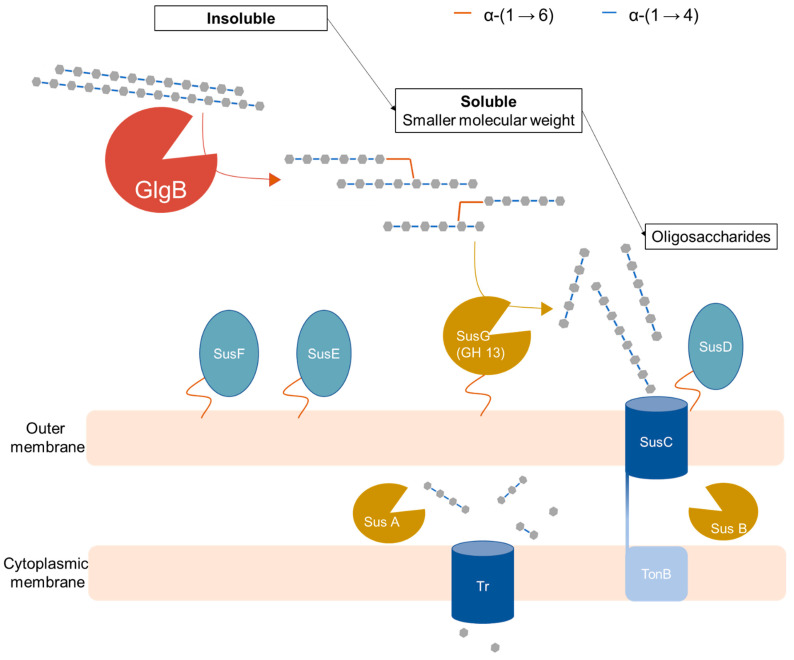
Schematic diagram of the putative role of glucan-branching enzyme (GlgB) in colonic starch degradation with the starch utilization system (Sus) of Gram-negative *Bacteroides*. The glucan-branching enzyme (GlgB) solubilizes amylose into branched structure (this study) that can be subsequently degraded by *Bacteroides* amylolytic system [1,2,10,11,12,31,32]. Sus G: extracellular amylase (neopullulanse); Sus D, E & F: protein involved in binding starch molecules; Sus C: TonB-dependent transporter (TBDT) that transport oligosaccharides molecules into periplasm; Sus A & B: periplasmic amylolytic enzymes (GH13 & GH97); Tr: Transporter that transports small saccharides into cytoplasm; TonB: cytoplasmic transmembrane complex.

**Table 1 molecules-28-01881-t001:** Concentration of iodine-binding amylose in different starches and amylose before and after the treatment with GlgB. Data are shown as average ± standard deviation of three replicate experiments.

	Amylose Content (g/L)	
	Control	GlgB Treatment
Amylose	7.65 ± 0.60	0.23 ± 0.05
Pea	4.53 ± 0.31	0.14 ± 0.03
Fava bean	4.37 ± 0.14	0.14 ± 0.03
Potato	3.91 ± 0.16	0.12 ± 0.02
Corn	3.56 ± 0.28	0.44 ± 0.07
Wheat	3.21 ± 0.31	0.31 ± 0.04
Barley	2.98 ± 0.25	0.46 ± 0.02

**Table 2 molecules-28-01881-t002:** Concentration of reducing ends in different starches and amylose before and after treatment with GlgB, after debranching by isoamylase and pullulanase. Data are shown as average ± standard deviation of two replicate experiments. Values for GlgB-treated starches are marked with an asterisk if treatment significantly (*p* < 0.05) increased the number of reducing ends.

	Reducing Ends μM/g Starch	
	Control	GlgB Treatment
Amylose	71.4 ± 3.6	239 ± 11 *
Pea starch	223 ± 5.7	250 ± 15
Fava bean starch	215 ± 2.1	248 ± 10 *
Potato starch	239 ± 10	252 ± 15
Corn starch	236 ± 8.2	258 ± 11 *
Wheat starch	229 ± 1.5	249 ± 18
Barley starch	234 ± 0.1	251 ± 14

**Table 3 molecules-28-01881-t003:** Concentration of glucose released after digestion from different starches after hydrolysis with pancreatic amylases and brush border enzymes. Starches were dissolved at a concentration of 10 g/L and treated with GlgB, or not. Data are shown as average ± standard deviation of three replicate experiments. Values for GlgB-treated starches are marked with an asterisk if treatment significantly (*p* < 0.05) reduced the glucose release from starch during in vitro enzymatic digestion.

	Glucose Release (g/L)	
	Control	GlgB Treatment
Pea starch	5.06 ± 0.31	4.44 ± 0.23 *
Fava bean starch	5.10 ± 0.44	4.25 ± 0.47 *
Potato starch	5.35 ± 0.33	4.87 ± 0.24
Corn starch	4.96 ± 0.78	4.48 ± 0.57
Wheat starch	5.05 ± 0.42	4.40 ± 0.45 *
Barley starch	5.35 ± 0.55	4.69 ± 0.25

## Data Availability

Data are contained within the article and its Appendix A.

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
