# Peer review of "Characterization of the Glucan-Branching Enzyme GlgB Gene from Swine Intestinal Bacteria"

_molecules, 2023, doi:10.3390/molecules28041881_

Round 1
Reviewer 1 Report
I have read the paper, and in principle the experimental work is sound, and the manuscript is well written.
I have however a serious concern with the discussion of the manuscript. My concern is that the Glucan-branching enzyme GlgB is described as an extracellular enzyme. However, it is generally accepted that in microorganisms the function of glucan branching enzymes is the synthesis of the branches in glycogen synthesis, which is intracellular process. As far as I know all glucan branching enzymes are intracellular enzymes. In this manuscript, however, it is hypothesised that the enzyme plays an important role in the initial steps of starch degradation in the intestine, before the starch becomes available to be used by the microbiota.
Importantly, no evidence is provided that it is an extracellular enzyme. I tried to analyse the sequence, but it was not yet available in Genbank. Please add this data. Therefore, I checked the Butyrivibrio fibrisolvens GBE sequence instead referred to in the materials and methods (98.5% identity) using freely available SignalP software, which indicates that it is intracellular enzyme. This makes it rather likely that the GBE of the study is also an intracellular enzyme. And this then contradicts the hypothesis made in the discussion.
The authors also refer to previous work of them, Wang et al., 2019, which would indicate the GBE to be an extracellular enzyme. And while many GH13 enzymes are extracellular enzymes (e.g. the starch degrading alpha-amylases), this is as far as I know not the case with the GH13 branching enzymes. It seems that by accident the authors of the current study concluded that also other GH13 enzymes extracellular as are alpha-amylases (found in their previous metagenomics study), which is not case.
Author Response
See attached file for our response

Reviewer 2 Report
The authors described the enzymatic characterization of Swine Intestinal bacteria GlgB on different starch based substrates and speculate on its modus operandi . The paper is brief, concised and overall well written. The aim is clear and execution of the expression of GlgB seems well done.
I have however several concerns regarding the paper:
1- The choice of substrates used in the manuscript. There is a single amylose sample and several starch from different origins. It is really hard to draw conclusion regarding the preference of GlgB on amylose. If this was the conclusions to be drawn then probably starches with different amylose contents (there are high amylose starches to waxy starches) that should have been a much more compelling story. Different branched structure addition would have been useful (for ex: Beta limit dextrin, glycogen...)
In addition, why using a mix of commercial amylose/potato starch vs lab purified starches? The purification process between commercial vs lab made is very different and may have an impact on the starch structure. That is also another weak point of the analysis to me.
If possible i would strongly recommend adding more starch (ideally commercial from the same company, from a unique variety and different amylose content. For example: potato, pure amylose, high amylose starch, standard starch, waxy starch...
2- The authors should add the % reduction of amylsoe in their table 1 and revise the comments and analysis of the table. I made the calculation and if I'm not mistaken Amylose, Pea (96.9), Fava bean (96.7) and potato (96.9) have a 97% reduction or close which does not match the analysis and conclusion in the text. The amylose assay is quite inaccurate in essence ... I don't think it is fair to say that the amylose content reduction was the highest with the amylose.
- Figure 3 it would have been useful to add arrows on where to look and if possible add standard as reference. all the profiles are strikingly different and hard to follow for the non initiates.
-Figure 3-4 are showing different subsamples (Amylose and starch from potato (fig 3)- potato starch vs whet starch (fig 4)... this makes really hard to follow and for clarity and transparency, it would have been preferable to show them all.
-Table 2: it seems to me that the measurement of reducing ends plateaued around 245-250 uM/uL. could it be the isoamylase/pullulanse half life? does the authors have checked adding some extra fresh enzyme aft12 hours in all to see if the value increase?
Minor comments:
- Figure 3 windows need to be annotated (A and B)
- discussion around usage of enzyme in the baking industry is an interesting point. while the use of GlgB could provide an interesting alternative, the current trend is to have a clean label product (meaning no additive). in addition, the cost of enzyme is the second highest in the list of bake good producers. It is true that high amylose can impair some baking properties but has been easily sorted by blending flours (which is what all baker do).
this is just a comment and not a requirement for any changes in the discussion.
Round 2
Reviewer 1 Report
Good improvements now, interesting paper.
Reviewer 2 Report
The authors have amended or discussed my points appropriately.
I'm happy for the journal to proceed.
Congratulations !!!